# Molecular Cloning and Characterization of a Serotonin *N*-Acetyltransferase Gene, *xoSNAT3*, from *Xanthomonas oryzae* pv. *oryzae*

**DOI:** 10.3390/ijerph20031865

**Published:** 2023-01-19

**Authors:** Xian Chen, Yancun Zhao, Pedro Laborda, Yong Yang, Fengquan Liu

**Affiliations:** 1State Key Laboratory for Managing Biotic and Chemical Treats to the Quality and Safety of Agro-Products, Key Laboratory of Biotechnology in Plant Protection, Ministry of Agriculture and Rural Affairs, Zhejiang Provincial Key Laboratory of Biotechnology in Plant Protection, Institute of Virology and Biotechnology, Zhejiang Academy of Agricultural Science, Hangzhou 310021, China; 2Institute of Plant Protection, Jiangsu Academy of Agricultural Sciences, Jiangsu Key Laboratory for Food Quality and Safety—State Key Laboratory Cultivation Base of Ministry of Science and Technology, Nanjing 210014, China; 3School of Life Sciences, Nantong University, Nantong 226001, China

**Keywords:** melatonin biosynthesis, *Xoo*, *Oryza sativa*, *xoSNAT3*, serotonin *N*-acetyltransferase

## Abstract

Rice bacterial blight (BB), caused by *Xanthomonas oryzae* pv. *oryzae* (*Xoo*), is one of the top ten bacterial plant diseases worldwide. Serotonin *N*-acetyltransferase (SNAT) is one of the key rate-limiting enzymes in melatonin (MT) biosynthesis. However, its function in pathogenic bacteria remains unclear. In this study, a *Xoo* SNAT protein (xoSNAT3) that showed 27.39% homology with sheep SNAT was identified from a collection of 24 members of GCN5-related *N*-acetyltransferase (GNAT) superfamily in *Xoo*. This xoSNAT3 could be induced by MT. In tobacco-based transient expression system, xoSNAT3 was found localized on mitochondria. In vitro studies indicated that xoSNAT3 showed the optima enzymatic activity at 50 °C. The recombinant enzyme showed K_m_ and *V_max_* values of 709.98 μM and 2.21 nmol/min/mg protein, respectively. Mutant △*xoSNAT3* showed greater impaired MT biosynthesis than the wild-type strain. Additionally, △*xoSNAT3* showed 14.06% less virulence and 26.07% less biofilm formation. Collectively, our results indicated that *xoSNAT3* services as a SNAT involved in MT biosynthesis and pathogenicity in *Xoo*.

## 1. Introduction

Melatonin (*N*-acetyl-5-methoxytryptamine, MT) is an evolutionary ancient and ubiquitous molecule present in animals, plants and microorganisms [1]. MT was first identified in animals in 1958 [2]. Thirty-six years later, MT was also identified in both plant and bacteria kingdoms [3,4]. As a well-known hormone, MT has been thoroughly studied in humans [5]. MT is synthesized and secreted by pineal gland in vertebrates, and its changes are related to the circadian rhythm, aging and immunity [6,7,8,9]. MT is a super-radical scavenger, not only because of the its structural characteristics, but also because MT has the ability to regulate the activity of antioxidant enzymes [10], and its antioxidant effects have been reported to be a beneficial treatment for COVID-19 patients [11]. MT also plays a key role as a master regulator in plant growth and stress responses [7]. It has been speculated that MT may act as a hormone in plants, and the first MT receptor has been recently identified in *Arabidopsis* [8]. Despite mentioned advances in animals and plants, the number of studies regarding the biological roles of MT in bacteria are limited and, thus, its function in bacteria is still poorly understood.

Systematic analysis of MT metabolic network may help to achieve a better understanding of its biological function. In vertebrates, MT is synthesized from tryptophan following four consecutive enzymatic steps [9]. Firstly, tryptophan is converted to 5-hydroxy tryptophan by the tryptophan hydroxylase (TPH). Then, this intermediate is converted to serotonin via an aromatic amino acid decarboxylase (AADC). Next, serotonin is further converted to *N*-acetylserotonin by the serotonin *N*-acetyltransferase (SNAT). Finally, *N*-acetylserotonin is converted to MT by the caffeic acid *O*-methyltransferase (COMT), also known as *N*-acetylserotonin *O*-methyltransferase (ASMT). Previous studies have reported that SNAT plays a critical role in the regulation of MT’s biosynthesis [10,11]. In many vertebrates, SNAT is localized in mitochondria and is believed to be the key rate-limiting enzyme in MT’s biosynthesis [12].

In plants, the biosynthesis of MT is quite different from that in vertebrates. Firstly, tryptophan is converted to tryptamine by the tryptophan decarboxylase (TDC) [13]. In the second step, this intermediate is converted to serotonin by the tryptophan 5-hydroxylase (T5H). Next, serotonin is *O*-methylated to 5-methoxytryptamine (5-MT) by ASMT/COMT. Finally, 5-MT is *N*-acetylated to MT by SNAT [14]. SNAT was reported to be localized in the chloroplasts, while T5H was located in the endoplasmic reticulum (ER), and TPH, ASMT/COMT and AADC/TDC were distributed in the cytoplasm [14,15]. TDC is also one of the key rate-limiting enzyme in plant MT’s biosynthesis [16]. Overexpression of *OsTDC* significantly enhances the content of MT in rice plants [17]. Although the biosynthetic steps of MT’s biosynthesis in vertebrates and plants have been thoroughly studied, research on the bacterial MT biosynthetic pathway are very limited. The changes in MT’s concentration have been correlated to the bacterial circadian rhythm [18]. The synthetic intermediates of serotonin, 5-hydroxytryptophan and *N*-acetylserotonin were identified in *Pseudomonas fluorescens* RG11, an endophytic bacterium isolated from grapevine roots [19]. In the cyanobacterium *Synechocystis* sp. PCC 6803, SNAT was reported to be a thermo-tolerant enzyme with 56% homology compared to rice OsSNAT [20].

In previous studies, our group reported that exogenous MT had strong antibacterial activity against *Xanthomonas oryzae* pv. *oryzae* (*Xoo*), the causal agent of rice bacterial blight (BB) [21]. *Xoo* is considered one of the top ten most dangerous bacterial pathogen in plant pathology [22]. In this study, endogenous MT was detected in *Xoo*, and the predicted xoSNAT3 was proved to be involved in MT biosynthesis. Additionally, the role of xoSNAT3 in the *Xoo* pathogenicity was characterized.

## 2. Materials and Methods

### 2.1. General Information and Bacterial Strain

All the chemicals and reagents were used as received from commercial suppliers without further purification. *Xoo* strain PXO99 (P6) was grown in nutrient broth (NB) medium (5 g peptone, 1 g yeast extract, 3 g beef extract paste and 10 g sucrose, pH 7.0–7.2, in 1 L of distilled water) or NA agar medium (NB broth with 17 g agar) [23]. *E. coli* DH5α and BL21 were grown in lysogeny broth (LB) medium (5 g of yeast extract, 10 g of triptone and 10 g of sodium chloride, pH 7.0–7.2, in 1 L of distilled water) or LB solid medium (LB liquid medium with 20 g agar).

### 2.2. Identification of MT in Xoo

To detect MT in *Xoo*, the bacterial metabolites were extracted using the water-ethyl acetate method. Firstly, bacterial cells in NB medium were harvested and washed with ddH_2_O, and the cells was adjusted to OD_600_ = 0.8~1.0. Then, 2 mL of standard bacterial was added to fresh 200 mL NB broth medium and all cultures were incubated at 28 °C and 200 rpm. The cells were harvested after 24 h and washed with ddH_2_O twice. Then, the cells were re-suspended in 5 mL ddH_2_O. Next, suspension cells were mixed with an equal volume of ethyl acetate, and mixed vigorously on a vortex for 2 min. The mixture was centrifuged at 10,000 rpm for 3 min, and the supernatant (organic phase) was transfer to a new 15 mL tube and dried with a nitrogen evaporator with nitrogen flow. The dried samples were reconstituted in 200 μL methanol, and shaken in a vortex at 2000 rpm for 1 min. After centrifuging at 10,000 rpm for 3 min, the mixture was filtered through a 0.1 μm membrane and analyzed using a liquid chromatography instrument with a Time of Flight Mass Spectrometer (TOF, Applied Biosystems Sciex, triple TOF 5600). The metabolites were identified using a Liquid Chromatograph Mass Spectrometer (LC-MS/MS, Agilent Poroshell 120 EC-C18, 2.7 μm, 3.0 mm × 100 mm). Pure ddH_2_O containing 0.1% trifluoracetic acid and methanol were used as the A and B mobile phases, respectively. The gradient program was performed at a flow rate of 0.3 mL/min. All samples were performed in triplicate, and the results were presented as the mean ± standard deviation.

### 2.3. Phylogenetic Analysis of xoSNAT3 from Xoo Strain PXO99

In order to predict a *Xoo* gene encoding a serotonin *N*-acetyltransferase enzyme, the nine full-length proteins belonging to the GNAT family (Table 1), which were downloaded from the NCBI website (https://www.ncbi.nlm.nih.gov/ (accessed on 8 October 2022)), were performed to find SNAT homologs in *Xoo* strain PXO99A protein database (https://www.ncbi.nlm.nih.gov (accessed on 8 October 2022)). Amino acid sequences of xoSNAT1 (WP_027703221.1), xoSNAT2 (WP_011258206.1) and xoSNAT3 (WP_027703680.1) were download from NCBI. Multiple amino acid sequences alignments were performed using the MEGA 6 software (Version 6.0), based on the cloning of SNATs from rice plant, sheep and other species. The phylogenetic analysis was performed with MEGA 6 by using the neighbor joining method with 1000 bootstrap replicates.

### 2.4. Cloning of xoSNAT3 from Xoo Strain PXO99

To obtain the genomic DNA from *Xoo*, 0.5 mL of bacterial solution (OD_600_ = 1.0) was added to fresh 50 mL NB broth medium, and the culture was shaken at 28 °C and 200 rpm for 24 h. Then, 6 mL of *Xoo* was centrifuged at 10,000 rpm for 3 min, and the genomic DNA was isolated by using the TIANamp Bacteria DNA Kit (Cat. NO. DP302-02, Beijing, China). To amplify the *xoSNAT3* from PXO99, polymerase chain reaction (PCR) primers were designed based on the annotated sequence information of putative *N*-acyl-transferase (GenBank accession no. PXO_RS04110) using the forward primer 5′-ATGTCCACCACAGCCCT-3′, and reverse primer 5′- CAAAGGAGCCGCGCCGGCA-3′. The resulting PCR product was purified using the DNA Clean-Up kit (Cat. NO. CW2301M, CWBIO, Taizhou, China), and cloned into the PJET1.2 vector (Cat. NO. K1231, Thermo Fisher, Waltham, MA, USA). The map of PJET1.2 vector was available in addgene (https://www.addgene.org/124439/ (accessed on 8 October 2022)). The insert fragment was verified via sequencing analysis by Tsingke Biotechnology Co., Ltd. (Nanjing, China).

### 2.5. Subcellular Localization of xoSNAT3 in Tobacco Leaves

The full-length coding regions of *xoSNAT3* in PJET1.2 vector were amplified using restriction enzymes for the respective forward (*BamH*I) 5′-CGGGATCCATGTCCACCACAGCCCT-3′, and reverse primers (*Kpn*I) 5′-GGGGGTACCAGGAGCCGCGCCGGCA-3′. After digestion, the PCR product with restriction enzyme site were fused in frame with GFP in the PCV-eGFP–N_1_ vector [24]. Then, the sequenced plasmid was introduced into *Agrobacterium tumefaciens* EHA105, and then transferred to 3-weeks-old *Nicotiana benthamiana* leaves, using the transient expression method. Briefly, EHA105 cells with PCV-eGFP-*xoSNAT3* were harvested at 16 h post-inoculation. The harvest cells were then re-suspended in soaking solution [containing 10 mM MgCl_2_, 10 mM MES (pH 5.6), 200 μM acetylsyringone (As)], and the bacterial mixture was adjusted to OD_600_ = 0.8~1.0. EHA105 with PCV-eGFP-N_1_ vector was used as mock control. Finally, the mixtures were slowly penetrated into the back of *N*. *benthamiana* leaves. These tobacco plants were culture for 2 days at 25 °C. The GFP signal was excited at 395 nm and observed at 450–490 nm with LSM 710 (ZEISS, Jena, Germany) for confocal imaging.

### 2.6. Measurement of xoSNAT3 Enzymatic Activity In Vitro

The full-length coding regions of *xoSNAT3* in PJET1.2 was amplified using restriction enzymes with the respective the forward (*Bam*HI) 5′-CGGGATCCTCCACCACAGCCCT-3′, and reverse primers (*Not*I) 5′-GCGGCCGCAGGAGCCGCGCCGGCA-3′. After digestion, the PCR product with restriction enzyme site was fused in frame with GST in pGEX-6p-1 vector. Then, the sequenced plasmid was introduced into *E*. *coli* BL21. Then, the expression of GST-xoSNAT3 fusion protein was induced by 0.4 mM IPTG and examined by western blot. Next, GST-xoSNAT3 fusion protein was purified by GSTsep glutathione agarose resin, according to the manufacturer’s instructions. Then, the concentration of purified protein was measured using the Bradford method.

In order to further confirm that GST-xoSNAT3 was correctly and successfully expression in BL21 cells, 15 μg of GST-xoSNAT3 and GST proteins were separated on a 12.5% (*w/v*) SDS-PAGE gel overlaid with a 4% (*w/v*) sticking gel, respectively. Then, one gel was stained by Coomasie Brilliant Blue (CBB)R-250 and the rest one was electrotransferred onto a PVDF membrane. Next, the PVDF membrane was blocked in blocking buffer [20 mM Tris-HCl, pH 7.6, 0.5 mM NaCl, 0.005% (*v/v*) Tween 20, 1% (*w/v*) BSA] overnight, following by incubation with mouse anti-GST antibody (1:1000, Abmart, Shanghai, China). In addition, the PVDF membrane was incubated in Dylight 488 goat anti-mouse secondary antibody IgG (1:10,000, Abbkine, Carson, CA, USA). Binding of the Dylight 488 conjugated antibodies was detected on Fluorescence detector (ChemiDoc XRS, Bio-rad, Hercules, CA, USA).

For the enzymatic activity assay, GST-xoSNAT3 and GST were added to 100 μL reaction buffer (0.5 mM serotonin, 0.5 mM acetyl-CoA and 100 mM potassium phosphate, pH = 8.8), respectively. The mixture was incubated at 30 °C for 30 min, and the reaction was stopped by adding 25 μL methanol. Next, a 20 μL aliquot of the reaction samples was analyzed by HPLC to determine the substrates. To investigate the substrate affinity (K_m_) and *V_max_* values of xoSNAT3, different concentrations of xoSNAT3 were added to the above-described buffer, serotonin and *N*-acetylserotonin were both determined by LC-MS/MS. The mixture were filtered through a 0.1 μm membrane and analyzed by liquid chromatography in tandem with a Time of Flight Mass Spectrometer (TOF, Applied Biosystems Sciex, triple TOF 5600). The metabolites were identified using a Liquid Chromatograph Mass Spectrometer (LC-MS/MS, Agilent Poroshell 120 EC-C18, 2.7 μm, 3.0 mm × 100 mm). Pure ddH_2_O containing 0.1% trifluoracetic acid and methanol were used as the A and B mobile phases, respectively. The compounds were eluted at a flow rate of 0.3 mL/min. All samples were performed in triplicate, and the results were presented as the mean ± standard deviation.

### 2.7. Generation of xoSNAT3 Deletion Mutant

In *Xoo*, *xoSNAT3* (nucleotides 901096-901596) is located 251 bp downstream of PXO_RS04100 and 6 bp upstream of PXO_RS04115. To generate a nonpolar mutation in *xoSNAT*, fragments located 509 bp upstream and 510 bp downstream of *xoSNAT* were amplified from *Xoo* genomic DNA using the primers *snat up-F* (*Bam*HI): 5′-CGGGATCCTCACGCACGACGACGTGCG-3′, *snat up-R*: 5′-CATGCGAACTCCAAAGGAGGGTGGACATCACCGCATGA-3′, *snat D-F*: 5′-T CATGCGGTGATGTCCACCCTCCTTTGGAGTTCGCATG-3′, and *snat D-R* (*Xba*I): 5′-GCTCTAGACACCTGCGTACGGGTACGC-3′, respectively. Then, the upstream and downstream PCR productions were combined together using the PCR fusion method. Briefly, upstream (1 μL) and downstream (1 μL), 2× PCR Mix (10 μL, AS102-01, TRAN) and ddH_2_O (7.6 μL) were mixed together in a 200 μL PCR tube. The PCR mixture were performed on a S1000 PCR system (Bio-rad, Hercules, CA, USA), and PCR conditions were as follows: 98 °C for 3 min, then 10 cycles of 98 °C for 5 s, 60 °C for 10 s, 68 °C for 20 s, with a final 68 °C for 2 min. The resulting PCR product was cloned into vector pMDT18-T, and verified by DNA sequencing. The construct was digested with *Bam*HI and *Xba*I to release the cloned fragment, and then ligated into the vector pK18mob*sacB*. The recombinant plasmid was introduced into *Xoo* by electroporation. Briefly, 10 μL recombinant plasmid mixed with 100 μL *Xoo* cells, and placed on ice for 10 min. Then, this mixture was transfer into electric shock cup, and electric shock by electric shock apparatus (Bio-rad, MicroPulser, Hercules, CA, USA). A mutant of lacking *xoSNAT3* was initially obtained in NAN medium, and then on the NA medium following the procedure. Firstly, the transconjugants were selected on NA plates with kanamycin (Km, 50 μg/mL) in the absent of sucrose. Then, positive colonies were selected on NA plates with 10% (*w/v*) sucrose to generate the in-frame deletion mutant via allelic homologous recombination. The resulting mutant, containing the *xoSNAT* in-fame deletion, was further confirmed by PCR and quantitative RT-PCR. Three of the confirmed mutants, named △*xoSNAT3*, were selected for further study.

### 2.8. Complementation of the xoSNAT3 Mutant

For the complementation of *xoSNAT3*, a 489 bp DNA of the entire coding region of *snat* was amplified from the *Xoo* strain PXO99 genomic DNA using the forward (*EcoR*I): 5′-CGGAATTCATGTCCACCACAGCCCTCCCT-3′ and reverse primers (*Bam*HI): 5′-CGGGATCCCAAAGGAGCCGCGCCGGCAGG-3′. The PCR product was cloned into the vector pMDT18-T and verified by sequencing. The construct was digested with *EcoR*I and *BamH*I to release the PCR-fragment, and then ligated into the vector pUFR034. The complemented plasmid pUFR-sant was transformed into the competent cell of △*xoSNAT* by electroporation. Finally, one representative complemented strain, named △*xoSNAT*, was selected on NA plates with Km (50 μg/mL), verified by PCR and used in the subsequent studies.

### 2.9. RNA Extraction and Quantitative RT-PCR Analysis

Special primers for qRT-PCR of *xoSNAT* were designed using PRIMER 5 (v. 5) software: forward primer 5′-GTCCACCACAGCCCTCCCT-3,′ and reverse primer 5′-GTAGCTTTGCCGTCCAGTTCC-3.′ The housekeeping gene *16S rRNA* (forward primer: 5′-CAAGGCGCTGCTGATGGTCG-3;′ reverse primer: 5′-CGTCGCAAGATCGCGTTGACC-3′), and *recA* (forward primer: 5′-AATGCCTTGAAGTTCTACGCC-3,′ reverse primer: 5′-TTCGGTCACGACCTGCTTG-3′) were used as internal controls. To obtain the RNA from *Xoo*, △*xoSNAT* and its complementation, 0.5 mL of bacterial solutions (OD_600_ = 1.0) were added to fresh 50 mL NB and the culture was shaken at 28 °C and 200 rpm for 24 h. Then, 3–5 mL of bacterial suspension was centrifuged at 10,000 rpm for 3 min, and the total RNA was isolated using TRIzol reagent (Invitrogen, Carlsbad, CA, USA). Total RNA was treated with DNase I (Takara) to eliminate genomic DNA and the first strand cDNA was synthesized by using the cDNA Synthesis kit (Takara, Bio, Nojihigashi, Kusatsu, Japan) following the manufacturer’s instructions. The qRT-PCR was performed on a QuantStudio 6 Real-Time PCR system (Applied Biosystems, Waltham, MA, USA), using diluted cDNA and the SYBR Green PCR Master Mix (Takara, Nojihigashi, Kusatsu, Japan). The expression data, in terms of quantification cycles threshold (Ct), were collected and statistically processed using the 2^−△△Ct^ method. Each experiment was conducted three times with three replicates. The variables were analyzed via Student’s t test and tested for significance at *p* < 0.05, *p* < 0.01, *p* < 0.001 and *p* < 0.0001 levels.

### 2.10. Analysis of Pathogenicity and Biofilm Formation

Pathogenicity assays were performed in a glasshouse. *Xoo*, △*xoSNAT* and its complementation were cultivated in NB medium at 28 °C and 200 rpm for 24 h. The cells were collected and resuspended in sterilized ddH_2_O to OD_600_ = 1.0. For pathogenicity assay, the strains were inoculated into the leaves of 4- to 5-week-old rice plants (variety Nipponbare, which is susceptible to BB), using the leaf clipping method [25]. The lesion length was measured at 7 days post inoculation. Thirty leaves were treated with each strain. The experiments were conducted three times. The variables were analyzed via Student’s t test and tested for significance at *p* < 0.05, *p* < 0.01, *p* < 0.001 and *p* < 0.0001 levels.

Biofilm formation was measured as previously described [21,26]. Briefly, *Xoo*, △*xoSNAT* and its complementation were cultivated in NB medium with shaking at 28 °C and 200 rpm for 24 h. The cells were collected and resuspended in sterilized ddH_2_O to OD_600_ = 1.0. Next, 30 μL cell suspension was inoculated into 3 mL NB liquid broth medium and placed in darkness without shaking for 5 days. After gently removing the cultures, the cells adhered to the tubes were stained with crystal violet method and the absorbance of OD_595_ was measured using a spectrophotometer (Eppendorf Biophotometer Plus, Hamburg, Germany). Each experiment was performed three times, with seven replicates each time. The variables were analyzed via Student’s t test and tested for significance at *p* < 0.05, *p* < 0.01, *p* < 0.001, and *p* < 0.0001 levels.

### 2.11. Bioinformatics Analysis of xoSNAT3

In order to predict the function, theoretical pI (isoelectric point), MW (molecular weight) and subcellular location were calculated. The amino acid sequence of xoSNAT3 was blasted in Uniprot (https://www.uniprot.org/ (accessed on 8 October 2022)) and pfam (http://pfam.xfam.org/ (accessed on 8 October 2022)) databases to determine *xoSNAT3* function. Parameters of Uniprot were as follows: sequence type-“protein.” program-“blastp,” E-threshold-“10,” Matrix-“Auto Blosum62,” Filter-“None,” Gapped-“Yes,” Hits-“250,” and HSPs per hit-“All.” The pfam analysis did not require parameter settings. Expasy (https://web.expasy.org/compute_pi/ (accessed on 8 October 2022)) was used to predict its theoretical pI and MW, and PSORT II (http://psort.hgc.jp/form2.html (accessed on 8 October 2022)) used to predict xoSNAT3 subcellular location. These last software did not require any parameter settings.

## 3. Results

### 3.1. Identification of MT and MT’s Synthetic Intermediates in Xoo Extracts

In order to determine whether *Xoo* can synthesize MT, endogenous MT was extracted with methanol and analyzed by mass spectroscopy (Figure 1). When using standard MT, a main *m/z* peak was observed at 233.1405 Da, obtaining MS/MS fragments at 115.0606, 130.0725, 159.0765, 174.1007, 175.1033 and 216.1130 (Figure 1A, Table 2). Interestingly, the same peak at 233.1290 Da was also detected in the *Xoo* extract, observing similar MS/MS fragments (114.0909, 131.0726, 159.0675, 174.0915, 175.0946 and 216.1013) compared to those observed when using MT standard (Figure 1B). Thus, our results suggested that *Xoo* can produce MT.

To further investigate MT synthetic pathway in *Xoo*, the synthetic intermediates were extracted after 24 h of incubation and analyzed by LC-MS. When using standard compounds, the retention times for tryptophan, tryptamine, 5-hydroxytryptan, 5-hydroxytryptamine, *N*-acetylserotonin and MT were 7.844, 7.847, 5.154, 5.559, 8.131 and 9.801 min, respectively. In agreement, the compounds were detected in *Xoo* extracts at 7.850, 7.861, 5.160, 5.610, 8.130 and 9.800 min (Figure 2), respectively. The content of tryptophan, tryptamine, 5-hydroxytryptan, 5-hydroxytryptamine, *N*-acetylserotonin and MT in *Xoo* cells was 486 ± 2.6, 639 ± 2.3, 248 ± 1.9, 265 ± 1.1, 63 ± 1.5 and 66 ± 2.3 ng/mL, respectively (Appendix A).

### 3.2. Phylogenetic Analysis of xoSNAT3

It is well known that finding synthetic genes is the most difficult and crucial step in synthetic biology. The most common and effective way to identify proteins in different species is by homologous comparison. In order to identify the SNAT-ecnoding genes from *Xoo*, seven full-length proteins belonging to GNAT superfamily (pfam00583) (Table 1), were submitted to BLAST search in *Xoo* protein database (https://www.ncbi.nlm.nih.gov/ (accessed on 8 October 2022)). Fortunately, three potential xoSNATs proteins were found in *Xoo* (Figure 3A,B). Phylogenetic analysis showed that Sheep SNAT and OsSNATs are divided into two different subfamilies (Figure 3B). The protein sequences of the above three *Xoo* SNATs and SNATs protein from other species were presented in Figure 3A. Protein annotation showed that xoSNAT1 (WP_027703221.1), xoSNAT2 (WP_011258206.1), xoSNAT3 (WP_027703680.1) and Sheep SNAT (gi|11387097) belonged to the GNAT superfamily. Compared with xoSNAT1 and xoSNAT2, xoSNAT3 showed a closer evolutionary relationship compared with theSNATs from animal and plant kingdoms (Figure 3B). Sheep SNAT showed 32.36% (xoSNAT1), 25.15% (xoSNAT2) and 22.96% (xoSNAT3) homology, respectively. Homology analysis revealed that xoSNAT3 shared high identity with SaSNAT1 (48.56%), while the homology was only 31.09% and 27.37% when comparing SaSNAT1 with xoSNAT1 and xoSNAT2, respectively. This suggested that *xoSNAT3* may encode a SNAT protein.

### 3.3. Enzymatic Activity Analysis of xoSNAT3

The full length of xoSNAT3 contained 501 bp, thus encoding a protein with 166 amino acids. Protein molecular weight of xoSNAT3 and GST is 17.95 and 26 KDa, respectively. It was predicted that xoSNAT3 contained an acetyltransferase (GNAT) domain by Pfam and Uniprot database scanning. To confirm xoSNAT3 activity, its frame was fused into pGEX-6P-1 vector with *N*-terminal glutathione-*S*-transferase-tagging. After 12–16 h post IPTG induction, the purified SNAT3-GST fusion proteins were analyzed by SDS-PAGE and western blot, and its enzymatic activity and kinetic parameters were analyzed in vitro. As shown in Figure 4, GST-xoSNAT3 (43.95 KDa) and GST (26 KDa) proteins were successfully induced by IPTG, and further confirmed by western blotting (Figure 4A–C). The optima enzymatic activity of GST-xoSNAT3 was detected at 50 °C (Figure 4D). The values for K_m_ and *V_max_* using serotonin as the subtrate were 709.98 μM and 2.21 nmol/min/mg protein, respectively (Figure 4E).

### 3.4. Location of xoSNAT3 in Tobacco Cells

Protein functions are related to its subcellular localization [27]. It was predicted that xoSNAT3 had 60.9% likely to be located in mitochondria by PSORT II database scanning. To further investigate the subcellular localization of xoSNAT3, the recombinant 35: GFP plasmid with full length ORF (Opening Reading Frame) of *xoSNAT3* was transferred into tobacco leaves, and the GFP fluorescent signals were detected by confocal microscope. As shown in Figure 5, the GFP signal in the absence of xoSNAT3 was detected both in the plasma membrane and in the cell nucleus of tobacco tissue. The GFP fluorescent signals of xoSNAT3-GFP was observed only in mitochondria, similar with PSORTII result.

### 3.5. Role of xoSNAT3 in MT Biosynthesis in Xoo

To investigate whether *xoSNAT3* is induced in response to MT treatment, qRT-PCR was performed to analyze its transcription. The qRT-PCR analysis revealed that the mRNA level of *xoSNAT3* was higher after treatment with 1000 ng/mL MT, while its expression was down-regulated when treating with 100 ng/mL MT (Figure 6A). The results suggested that *xoSNAT3* can be induced when applying high MT concentrations.

To further examine the role of *xoSNAT3* in MT biosynthesis, the knock-out strain △*xoSNAT3* was constructed by a two-step homologous recombination approach. The mRNA of *xoSNAT3* was only detected in the wild-type strain PXO99 and in the complemented △*xoSNAT3*(*xoSNAT3*) strain, but not in △*xoSNAT3* (Figure 6B). To further investigate the relationship between MT biosynthesis and *xoSNAT3* gene expression, the concentration of MT was measured by LC-MS/MS in all strains. MT’s concentrations in the wild-type strain PXO99 and complemented strain △*xoSNAT3*(*xoSNAT3*) were 68.75 ng/50 mL and 99.25 ng/50 mL, respectively, whereas MT production level in △*xoSNAT3* mutant strain was only 3.75 ng/50 mL, which is consistent with the qRT-PCR analysis (Figure 6C). These results suggested that *xoSNAT3* is critical for MT biosynthesis in *Xoo*.

### 3.6. Role of xoSNAT3 in Xoo Pathogenicity

To examine the role of *xoSNAT3* in *Xoo* virulence, wild-type, △*xoSNAT3* mutant and complemented strain △*xoSNAT3*(*xoSNAT3*) were inoculated in rice Nipponbare leaves. As shown in Figure 7A, the lesion length produced by the wild-type and the complemented strain △*xoSNAT3*(*xoSNAT3*) were 7.97 and 7.48 cm, respectively. In contrast, the lesion length when using △*xoSNAT3* mutant was only 6.69 cm. Thus, *xoSNAT3* seems to be involved in the pathogenicity of *Xoo*.

Biofilm formation is crucial for bacterial colonization and virulence [26]. Biofilm-associated pathogens can form light-colored rings on the wall of a culture tube at the interface between air and broth. To further evaluate the effect of *xoSNAT3* on *Xoo* virulence, the biofilm formation of wild-type strain, △*xoSNAT3* mutant strain and complemented strain △*xoSNAT3*(*xoSNAT3*) were analyzed. As shown in Figure 7B, the crystal violet (CV) observation at OD_595_ of wild-type strain and complemented strain △*xoSNAT3*(*xoSNAT3*) were 0.45 and 0.47, respectively, while the CV value of △*xoSNAT3* mutant strain was only 0.33. This indicated that *xoSNAT3* is involved in biofilm formation in *Xoo*.

## 4. Discussion

Previous reports have indicated that MT provides a main *m/z* peak in positive mode at 233 [28,29,30], which is in agreement with the mass spectra obtained in this study. This confirms for the first time that MT is produced by *Xoo*, and probably by other *Xanthomonas* strains. The presence of MT in *Xoo* raises numerous questions regarding the potential role of this molecule in plant-*Xanthomonas* interactions. It is possible that *Xoo*-secreted MT is able to modify plant metabolism and may be related to *Xoo*-infection process. The role of MT in *Xoo* quorum sensing is also an important area that must be addressed in future studies.

MT synthetic routes in animals and plants have been well studied, while its synthetic pathway in microorganisms is still not well understood [9]. Tryptophan is the unique precursor of MT. Thus, the identification of intermediates may help to explore the bacterial MT biosynthetic pathway. In animals, the first step of MT synthesis pathway consists of the conversion of tryptophan into 5-hydroxytryptamine [9,31]. However, the first step of MT biosynthesis in the plant kingdom is the conversion of tryptophan into trypamine [14]. No study regarding the MT biosynthetic pathway in bacteria was reported until date. It has been reported to investigate MT intermediates by the N15 labelled tryptophan mediated isotopic tracer Method [24]. Interestingly, 5-hydroxytryptamine, instead of tryptamine, was detected in *Pseudomonas fluorescens* RG11, an endophytic bacterium isolated from grapevine roots. The 5-hydroxytryptamine is only consist in animal MT synthesis pathway. Therefore, the author speculates that the synthetic pathway of MT in RG11 may be similar to that reported in animals. Interestingly, tryptamine, 5-hydroxytryptamine, *N*-acetylserotonin and MT were also identified in *Xoo* by LC-MS/MS. Moreover, the highest content of tryptamine was 639 ng in 50 mL NA at 24 h post inoculation, while the lowest content of serotonin was 63 ng in 50 mL. It is well known that tryptamine is an intermediate of plant MT biosynthesis, whereas 5-hydroxytryptophan is involved in animal MT synthesis [14,18]. Surprisingly, tryptamine and 5-hydroxytryptophan were both detected in *Xoo* cells. However, no protein with high homology compared to reported TDCs was found in *Xoo* genome [18]. Further research is necessary in order to confirm if both pathways are present in *Xoo*.

Mature proteins must be correctly located in specific subcellular structures to perform their corresponding biological functions. Subcellular localization of rice OsSNAT indicated that this protein was located in chloroplast, while oocytes SNATs were localized in the mitochondria [15,32]. Our result suggested xoSNAT3 was located in mitochondria in tobacco leaves cells, which is a similar to that observed in animal cells. An important limitation in this study is that *Xoo* is a bacterium and, for this reason, does not contain mitochondria. Unfortunately, methods for confirming the location of proteins in *Xanthomonas* are lacking. The developed method using tobacco cells can be used in further experiments to examine co-localization of xoSNAT3 with bio-markers of plasma membrane, mitochondria and chloroplast, respectively. Despite the obtained results, further research is necessary to confirm the location of xoSNAT3 in bacterial cells.

The enzymatic activity of proteins is closely related to its biological function [33]. SNAT is the key rate-limiting enzyme that catalyzes the penultimate step in MT biosynthesis [34]. Sheep SNAT was identified in 1998 [35], while rice SNAT was cloned and characterized in 2013 [36]. OsSNAT showed K_m_ and *V_max_* values of 270 μgM and 3.3 nmol/min/mg protein, respectively; while the values of K_m_ and *V_max_* of the cSNAT from *Synechocystis* sp. *PCC 6803* were 823 μgM and 1.6 nmol/min/mg protein, respectively [15,20]. In this study, enzymatic activity assay indicated that GST and xoSNAT2 have no SNAT activity, while xoSNAT1 showed only low activity (Appendix A). The optimum temperatures of xoSNAT3 were similar to those observed when studying OsSNATs [15].

It is possible that *Xoo* has evolved to adapt to the same environmental conditions as rice during the epidemic season. Surprisingly, the *V_max_* of OsSNAT was 1.49 times higher than that of xoSNAT3. The obtained results revealed that xoSNAT3 is involved in *Xoo* pathogenicity and biofilm formation. However, further research is necessary to understand how MT is involved in these factors.

## 5. Conclusions

In this study, MT was firstly identified in *Xoo* by mass spectrometry. Our data showed that xoSNAT3 had SNAT activity in vitro. Subcellular localization using tobacco cells showed that oxSNAT3 was located in the mitochondria. Knocking out of *xoSNAT3* in *Xoo* strain PXO99 showed impaired MT production and reduced pathogenicity and biofilm formation than the wild-type strain. This study reveals for the first time the ability of *Xanthomonas* to synthesize MT, and provides new insights on the biological roles of MT in pathogenic bacteria.

## Figures and Tables

**Figure 1 ijerph-20-01865-f001:**
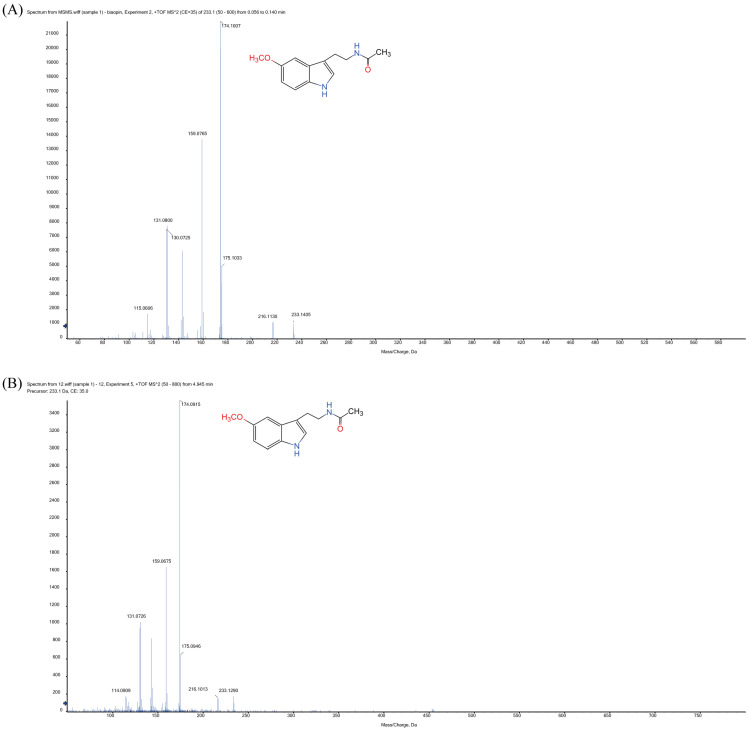
Detection of MT. (**A**) Mass spectrometry analysis of standard MT. (**B**) Mass spectrometry analysis of MT from *Xoo* extracts. *Xoo* cells were harvested after 24 h and the metabolites were extracted by ethyl acetate. The samples were reconstituted in 200 μL methanol and analyzed by mass spectrometry.

**Figure 2 ijerph-20-01865-f002:**
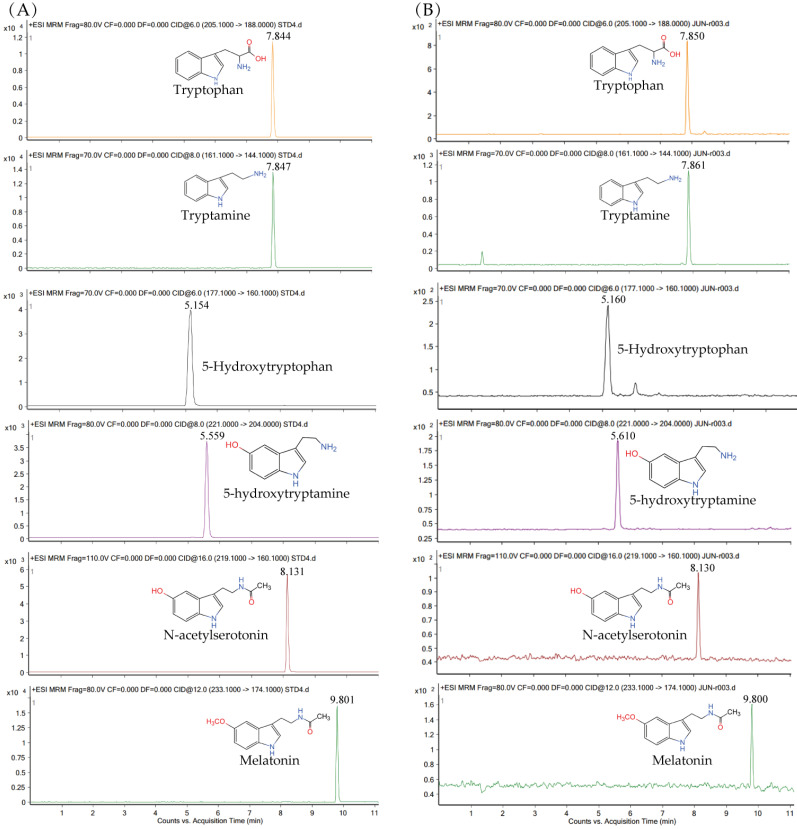
Detection of MT biosynthetic intermediates in *Xoo*. (**A**) Chemical structures and chromatograms of tryptophan, tryptamine, 5-hydroxytryptophan, 5-hydroxytryptamine, *N*-acetylserotonin and MT standards. (**B**) Chromatograms of tryptophan, tryptamine, 5-hydroxytryptamine, *N*-acetylserotonin and MT in *Xoo* cells. *Xoo* cells were harvested after 24 h and the metabolites were extracted with ethyl acetate. The dried samples were reconstituted in 200 μL methanol and analyzed by LC-MS/MS.

**Figure 3 ijerph-20-01865-f003:**
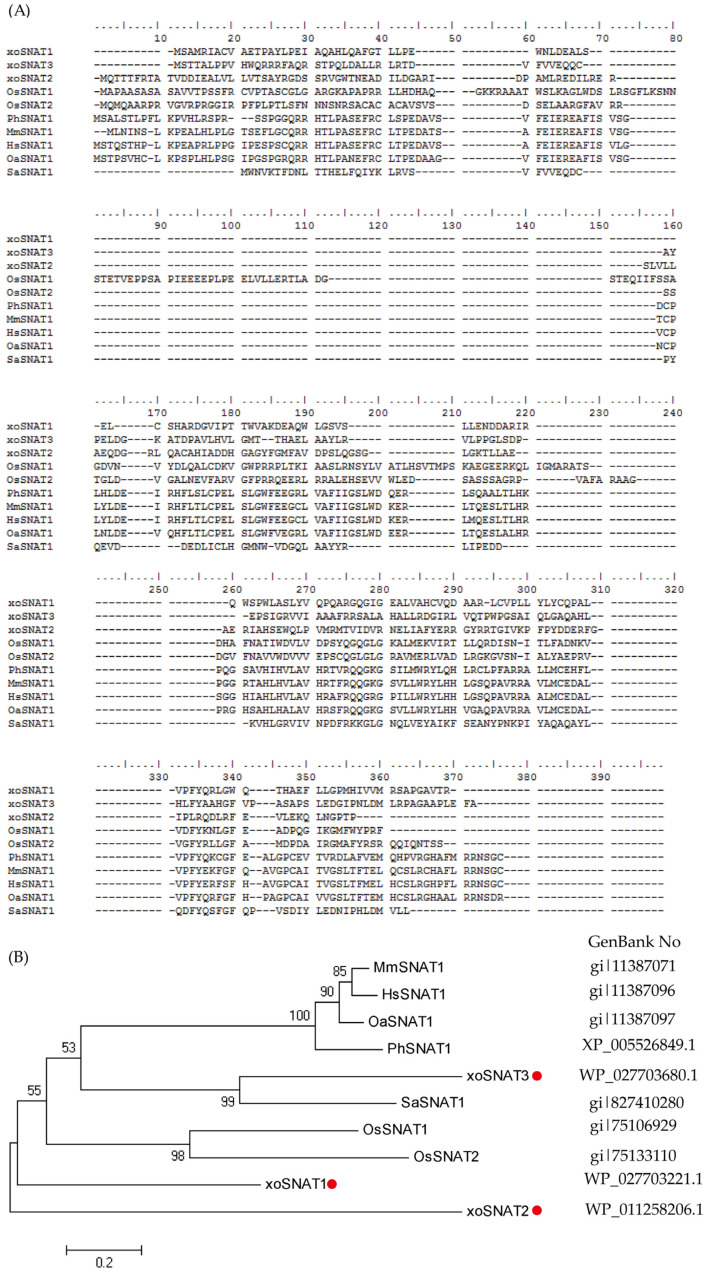
Phylogenetic tree of MT biosynthetic gene xoSNAT3. (**A**) Amino acid sequence alignment of xoSNAT3 and 9 GNAT family members. (**B**) Phylogenetic tree of xoSNAT3 and nine GNAT family members. The phylogenetic tree was constructed using MEGA-6, based on the amino acid sequences of SNATs from *Xoo* and some representative organisms. Phylogenetic analysis was performed by using Neighbor likelihood algorithm. The parameters were: Test of Phylogeny-“Bootstrap method,” No of Bootstrap replications-“1000,” Substitutions type-“Amino acid,” Rates among sites-“Uniform rates,” Pattern among Lineages-“Same (Homogeneous),” and Gaps/Missing Data Treatment-“Complete deletion”.

**Figure 4 ijerph-20-01865-f004:**
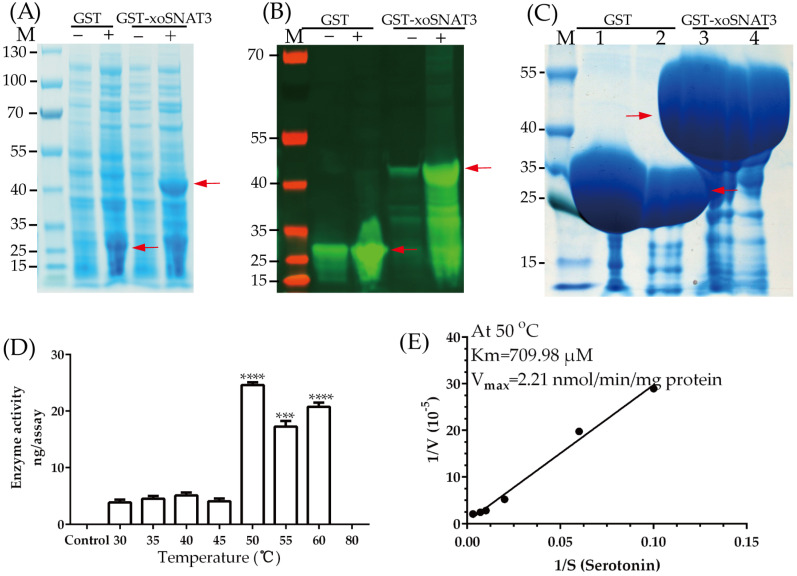
Characterization of the enzymatic activity of xoSNAT3. (**A**) Expression of GST-xoSNAT3 and GST tagged proteins in *E. coli* by IPTG. (**B**) Western blot analysis of purified GST-xoSNAT3 and GST. (**C**) Purification of GST-xoSNAT3 and GST. (**D**) Analysis of the enzymatic activity of GST-xoSNAT3 at different temperatures. (**E**) Determination of K_m_ and *V_max_* values of GST-xoSNAT3 using serotonin as the substrate. The expression of GST-xoSNAT3 in *E. coli* was induced by the addition of IPTG (0.4 mM) at 16 °C for 16 h. GST-xoSNAT3 (1 μg) was incubated with serotonin (0.5 mM) at different temperatures, the serotonin and product were both determined by LC-MS/MS. Control means enzymatic activity of 100 μL reaction buffer (0.5 mM serotonin, 0.5 mM acetyl-CoA and 100 mM potassium phosphate, pH = 8.8) in the absent of GST-xoSNAT3. The mixture was incubated at 30 °C for 30 min, and analyzed by HPLC. The K_m_ and *V_max_* values of GST-xoSNAT3 were determined by using Lineweaver-burk plots. Red arrow means the target protein. Asterisks indicate statistically significant differences determined using Student’s *t*-test (*** *p* < 0.001; **** *p* < 0.0001).

**Figure 5 ijerph-20-01865-f005:**
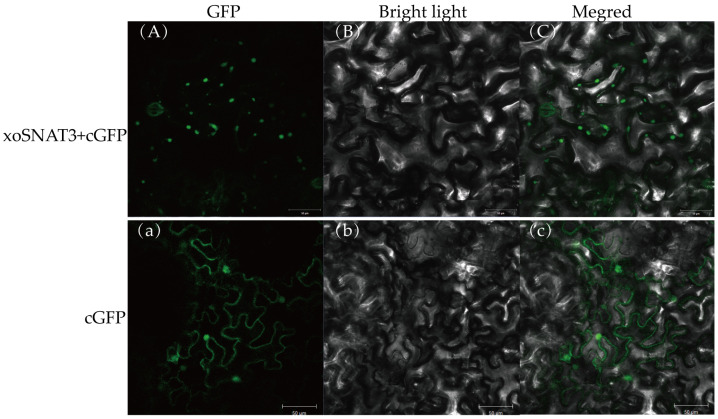
Localization of xoSNAT3 in tobacco cells. (**A**,**a**) Fluorescent images of xoSNAT3-GFP and GFP. (**B**,**b**) Bright Light filed of xoSNAT3-GFP and GFP. (**C**,**c**) Migration images of xoSNAT3-GFP and GFP. Recombinant plasmid with EHA105 was injected into 3-week-old tobacco leaves, and the GFP signal was observed by confocal imaging.

**Figure 6 ijerph-20-01865-f006:**
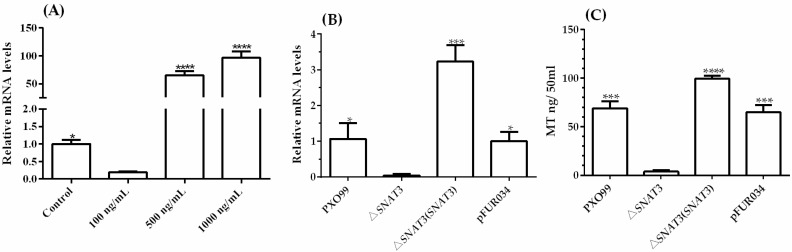
Expression of *xoSNAT3* in *Xoo* and under MT treatment. (**A**) The expression of *xoSNAT3* under MT treatment (100, 500 and 1000 ng/mL). (**B**) The expression of *xoSNAT3* in *Xoo*, △*xoSNAT3* mutant and complemented strain. (**C**) Concentration of MT in *Xoo*, △*xoSNAT3* mutant and complemented strain. The control experiment was carried out using *Xoo* strain PXO99 without MT treatment. PXO99 refers to the wild-type strain. pFUR034 refers to *Xoo* strain carrying mentioned blank plasmid as a mock control. Asterisks indicate statistically significant differences determined using Student’s t-test (* *p* < 0.05; *** *p* < 0.001; **** *p* < 0.0001).

**Figure 7 ijerph-20-01865-f007:**
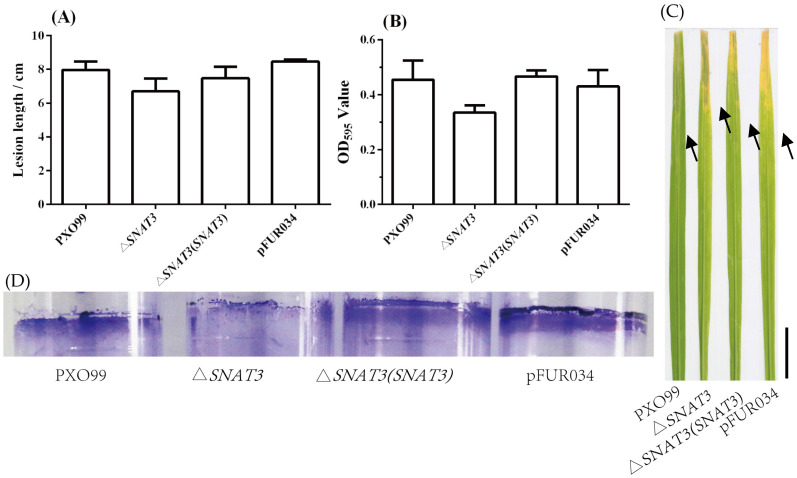
Characterization of the role of *xoSNAT3* in *Xoo* pathogenicity and biofilm formation. (**A**) Lesion length of *Xoo*, △*xoSNAT3* mutant and complemented strain in susceptible Nipponbare rice, at 7 days post-inoculation. (**B**) Biofilm formation by *Xoo*, △*xoSNAT3* mutant and complemented strain. (**C**) Images showing the symptoms produced by *Xoo*, △*xoSNAT3* mutant and complemented strains in rice leaves. (**D**) Images showing crystal violet staining of the biofilm formation by *Xoo*, △*xoSNAT3* mutant and complemented strain. pFUR034 refers to *Xoo* strain carrying mentioned blank plasmid as a mock control. Arrow refers to junction between disease and health areas. Bar = 3 cm.

**Table 1 ijerph-20-01865-t001:** List of ten full-length proteins information for GCN5-related *N*- acetyltransferases (GNAT).

No	Protein	Accession NO (NCBI)	AA Length	Functional Annotation	Specie
1	xoSNAT1	WP_027703221.1	161	*N*-acetyltransferase	*Xanthomonas* (Microorganism)
2	xoSNAT2	WP_011258206.1	179	*N*-acetyltransferase	*Xanthomonas* (Microorganism)
3	xoSNAT3	WP_027703680.1	166	*N*-acetyltransferase	*Xanthomonas* (Microorganism)
4	OsSNAT1	gi|75106929	254	Serotonin *N*-acetyltransferase	*Oryza sativa* (Plant)
5	OsSNAT2	gi|75133110	200	Serotonin *N*-acetyltransferase	*Oryza sativa* (Plant)
6	PhSNAT1	XP_005526849.1	205	Serotonin *N*-acetyltransferase	*Pseudopodoces humilis* (Animal)
7	MmSNAT1	gi|11387071	205	Serotonin *N*-acetyltransferase	*Mus musculus* (Animal)
8	HsSNAT1	gi|11387096	207	Serotonin *N*-acetyltransferase	*Homo sapiens* (Animal)
9	OaSNAT1	gi|11387097	207	Serotonin *N*-acetyltransferase	*Ovis aries* (Animal)
10	SaSNAT1	gi|827410280	142	*N*-acetyltransferase	*Streptococcus agalactiae* (Microorganism)

AA: Amino Acid sequence.

**Table 2 ijerph-20-01865-t002:** Top 7 peaks of MS/MS fragments from standard MT and MT from *Xoo* extracts.

NO	Standard MT	MT from *Xoo* Extracts
1	174.1007	174.0915
2	159.0765	159.0675
3	131.0800	131.0726
4	175.1033	175.0946
5	115.0606	114.0909
6	233.1405	233.1290
7	216.1130	216.1013

## Data Availability

Not applicable.

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
