# Peer review of "Molecular Cloning and Characterization of a Serotonin N-Acetyltransferase Gene, xoSNAT3, from Xanthomonas oryzae pv. oryzae"

_ijerph, 2023, doi:10.3390/ijerph20031865_

Round 1
Reviewer 1 Report (New Reviewer)
The focus of this work is trying to identify a serotonin N-acetyltransferase (SNAT), xoSNAT3, from pathogenic bacteria Xanthomonas oryzae pv. oryzase (Xoo). This paper detected endogenous melatonin (MT) with mass spectrometry in Xoo for the first time, characterized SNAT activity of xoSNAT3, and improved understanding of MT synthetic pathway in microorganisms. This paper also gave a hint that xoSNAT3 was related to Xoo infection process and pathogenicity to plant.
This study represents an important advance of the understanding of melatonin synthetic pathway in microorganisms. This study would be of much wider interest if provide much more details of enzymatic activity assay in vitro.
1. To detect melatonin in Xoo with mass spectrometry, did the authors purify Xoo metabolites with HPLC firstly? Compared melatonin only, it’s so clean for Xoo metabolites. If not, please provide the whole mass spectrometry results.
2. In line 160, when expressing protein with GST in pGEX-6p-1 vector, it’s better to name resulting protein to GST-xoSNAT3, because in fusion protein, GST is in N terminal.
3. In panel B of figure 2, it’s a little clean for Xoo extracts in mass spectrometry results. Please provide the whole LC-MS results.
4. In line 347, ‘pGEX6P1’ should be ‘pGEX-6p-1’.
5. For the enzymatic activity assay, it would be helpful for author to detect the differences of compounds before and after the reaction with LC-MS.
6. In panel D of figure 4, the enzymatic activity increased at a higher temperature, it would be useful to provide enzymatic activity at 60 -100 ℃. The enzyme should be inactivity at a enough high temperature.
7. In line 161, to express GST-xoSNAT3, IPTG concentration is 0.4 mM, while IPTG concentration is 0.6 mM in line 361. It would be helpful to double check this number.
8. It’s interesting for authors to investigate subcellular localization of xoSNAT3 in tobacco cells, not bacteria Xoo. To detect subcellular localization of xoSNAT3 in tobacco cells, it’s useful to provide co-localization results with plasma membrane, mitochondria and chloroplast markers.
9. In panel C of figure 7, it would be helpful to explain what’s meaning of the arrow.
10. In reaction of xoSNAT3, acetyl-CoA, after reaction, became CoA with thiol (-SH) group. The free thiol of CoA can react with 7-diethylamino-3-(4′-maleimidylphenyl)-4-methylcoumarin (CPM), generating fluorescent product upon reaction. The authors should measure enzymatic activity of xoSNAT3 with different methods.
Author Response
To reviewer 1
Thanks for reviewing this MS in your busy schedule, and your valuable comments have greatly improved the level and quality of this MS.
Comments and Suggestions for Authors
The focus of this work is trying to identify a serotonin N-acetyltransferase (SNAT), xoSNAT3, from pathogenic bacteria Xanthomonas oryzae pv. oryzase (Xoo). This paper detected endogenous melatonin (MT) with mass spectrometry in Xoo for the first time, characterized SNAT activity of xoSNAT3, and improved understanding of MT synthetic pathway in microorganisms. This paper also gave a hint that xoSNAT3 was related to Xoo infection process and pathogenicity to plant.
This study represents an important advance of the understanding of melatonin synthetic pathway in microorganisms. This study would be of much wider interest if provide much more details of enzymatic activity assay in vitro.
- To detect melatonin in Xoo with mass spectrometry, did the authors purify Xoo metabolites with HPLC firstly? Compared melatonin only, it’s so clean for Xoo metabolites. If not, please provide the whole mass spectrometry results.
Response:
Following the suggestion, we have provided the whole mass spectrometra.
This was indicated in lines 294 (Figure 1).
- In panel B of figure 2, it’s a little clean for Xoo extracts in mass spectrometry results. Please provide the whole LC-MS results.
Response:
Response:
Thanks for your kind suggestion. We have provides the whole LC-MS results as Figure S1.
Mentioned Figure was linked in lines 316-319 (Figure S1).
- For the enzymatic activity assay, it would be helpful for author to detect the differences of compounds before and after the reaction with LC-MS.
- In panel D of figure 4, the enzymatic activity increased at a higher temperature, it would be useful to provide enzymatic activity at 60 -100 ℃. The enzyme should be inactivity at a enough high temperature.
- In reaction of xoSNAT3, acetyl-CoA, after reaction, became CoA with thiol (-SH) group. The free thiol of CoA can react with 7-diethylamino-3-(4′-maleimidylphenyl)-4-methylcoumarin (CPM), generating fluorescent product upon reaction. The authors should measure enzymatic activity of xoSNAT3 with different methods.
Response:
Thanks for your kind suggestion. We have made changes following your suggestions. We have detected the reaction in the absence of xoSNAT3, and temperature-dependency with inactivated xoSNAT3.
Mentioned changes were added in lines 362-373.
- It’s interesting for authors to investigate subcellular localization of xoSNAT3 in tobacco cells, not bacteria Xoo. To detect subcellular localization of xoSNAT3 in tobacco cells, it’s useful to provide co-localization results with plasma membrane, mitochondria and chloroplast markers.
Response:
Thanks for your kind suggestion. We have made changes following your suggestion.
Changes were added in lines 474-477.
Minor mistakes
- In line 160, when expressing protein with GST in pGEX-6p-1 vector, it’s better to name resulting protein to GST-xoSNAT3, because in fusion protein, GST is in N terminal.
- In line 347, ‘pGEX6P1’ should be ‘pGEX-6p-1’.
- In line 161, to express GST-xoSNAT3, IPTG concentration is 0.4 mM, while IPTG concentration is 0.6 mM in line 361. It would be helpful to double check this number.
- In panel C of figure 7, it would be helpful to explain what’s meaning of the arrow.
Response:
Thanks for your kind suggestion. We have made changes following your suggestions

Reviewer 2 Report (New Reviewer)
The manuscript “Molecular Cloning and Characterization of a Serotonin N-Acetyltransferase Gene, xoSNAT3, from Xanthomonas oryzae pv. oryzae” can be published after some minor corrections.
Some suggested minor revisions:
Line 26- 50 ºC - check the presentation of degrees centigrade
Line 45 – give a space between Arabidopsis and [8].
Line 73 – give a space between Synechocystis sp. and PCC
Line 77 – after Xanthomonas oryzae pv. Oryzae put Xoo in brackets – (Xoo)
Line 122 – put “Table 1.”, and in Table 1, you should put the name of the species in italics
Line 136 – give a space between addgene and (https://www.addgene.org/124439/).
Line 170 - v/v in brackets – (v/v)
Line 258 - remove comma - “strain.,”
Line 318 – table in capital letters “Table”
Line 442 – give a space, “into5-hydroxytryptamine” to “into 5-hydroxytryptamine”
in the document, xoSNAT3 is sometimes in italic, sometimes not. You need to review.
Author Response
To reviewer 2
Thanks for reviewing this MS in your busy schedule, and your valuable comments have greatly improved the level and quality of this MS.
Minor mistake
Line 26- 50 ºC - check the presentation of degrees centigrade
Line 45 – give a space between Arabidopsis and [8].
Line 73 – give a space between Synechocystis sp. and PCC
Line 77 – after Xanthomonas oryzae pv. Oryzae put Xoo in brackets – (Xoo)
Line 122 – put “Table 1.”, and in Table 1, you should put the name of the species in italics
Line 136 – give a space between addgene and (https://www.addgene.org/124439/).
Line 170 - v/v in brackets – (v/v)
Line 258 - remove comma - “strain.,”
Line 318 – table in capital letters “Table”
Line 442 – give a space, “into5-hydroxytryptamine” to “into 5-hydroxytryptamine”
Response:
Thanks for your suggestions. We have revised the minor mistakes as indicated.
in the document, xoSNAT3 is sometimes in italic, sometimes not. You need to review.
Response:
Thanks for your suggestion.
When xoSNAT3 refers to gene, it is written in italics; while xoSNAT3 refers to protein, it is not written in italics. We have carefully checked itthroughout the MS.

Round 2
Reviewer 1 Report (New Reviewer)
The authors have answered all my questions. I have no more comments.
This manuscript is a resubmission of an earlier submission. The following is a list of the peer review reports and author responses from that submission.
Round 1
Reviewer 1 Report
Some observations to authors, included some mistakes and phrases and paragraphs less comprehensible.
ABSTRACT
Line 18: “…a SNAT related protein…”
INTRODUCTION
Line 41: “As in the case of vertebrates, MT also plays a key role as a master regulator in 41 plant growth and stress responses [12]”. The phrase is unclear, plants are not vertebrades. Omit: As in the case of vertebrates.
I suggest summarize information related to vertebrates and highlight information related to plant-microorganism interaction. MT is involved in plant growth regulation, evidence suggests that Xoo produces MT. Therefore, Xoo produce MT to manipulate plant metabolism? Why? or MT is a bacterial modulator? Any relationship with Quorum sensing?
MATERIALS AND METHODS
Line 191: “…(EcoRI):…”
Line 216: “ … using 2-ΔΔCt method…”
Line 220: “…on 50 mL NB. All…”
RESULTS
Line 250: “…Figure 1A). Interestingly…”
Line 277. The concentration of intermediates did not showed statistical differences (Figure S1).
Line 282. Table 1 is absent
Line 284: “…Phylogenetic analysis…”
Line 292: the figure 3 corresponds to phylogenetic reconstruction, however it is difficult to understand due to the complex terms. The Xoo sequences should be highlighted.
Information about sequence alignment and nucleotide substitution model is absent.
Section 3.2 “xoSNAT3 is the first reported SNAT from Xanthomonas” is difficult however it is difficult to understand due to the complex terms. I suggest author should use simplified terms instead codes from NCBI.
Section “3.3. xoSNAT3 is probably located in Xoo mitochondria” is not comprehensible. Since bacteria lack mitochondria. The authors should revise the results and rewriting.
Line 323: What kind of cells? Author should be specific.
Figure 6. Letters of statistical significance among treatments are absent.
Figure 7. I suggest authors should include representative photographs of damaged tissue and biofilms. Letters of statistical significance among treatments are absent.
Sections 3.2 and 3.2 are problematic since I did not perceive continuity between such sections. The phylogenetic reconstruction is not clear. The transient expression is not clear. The employed terms are difficult to follow. I suggest the authors would rewrite the sections.
DISCUSSION.
The authors speculate that biosynthesis of MT in Xoo may be similar to vertebrates. However, additional evidences are not shown. I suggest authors discuss about MT biosynthesis in diverse prokaryotes, including both pathogens and no pathogens.
The phrase “Our result suggested xoSNAT3 was located in mitochondria, which is a similar to that observed in animal cells. However, whether MT biosynthesis pathway in Xoo is similar to the one in animal cells needs further exploration” is very problematic. I did not understand the respective interpretation.
Line 406. The figure S3 is the first time was mentioned. The information is unclear.
CONCLUSION
The evidence suggests that Xoo produces MT as a part of pathogenicity, the host rice produces MT. However, a discussion about the relationship about MT biosynthesis considering the plant-pathogen interaction is necessary. Also a discussion about similarities between biosynthetic pathways from eukaryotes and prokaryotes is required. In my own perspective, the proposed model for biosynthesis of MT in Xoo is not well supported by current experimental evidence.
The evidence about the localization of xoSNAT was not considered in conclusion.
The manuscript includes figures but not tables. Information and discussion about Figures S1 and S2 is absent.
Author Response
Dear Editor,
We are resubmitting the manuscript titled "Molecular cloning and characterization of a serotonin N-acetyltransferase gene, xoSNAT3, from Xanthomonas oryzae pv. oryzae" to international Journal of Environmental Research and Public Health, in the section of Environmental Health, in the special Issue "New Strategies for the Environment-Friendly Management of Plant Diseases.
Comments from reviewers of the previous submission were greatly appreciated, and our manuscript has been modified accordingly. Attached are the response to reviewers.
Rice bacterial blight caused by Xoo is a major rice disease and serves as a model system for studying disease resistance in plants. The molecular mechanisms underlying the direct effect of melatonin on plant pathogenic are very complex. Serotonin N-Acetyltransferase (SNAT) is one of the key rate-limiting enzymes in melatonin (MT) biosynthesis. Here, we have identified SNAT related protein (xoSNAT3) in Xoo. xoSNAT3 was localized on mitochondria and has serotonin catalytic activity. Knock out xoSNAT3 impaired MT biosynthesis, and showed 14.06% less virulence and 26.07% less biofilm formation than the wild type strain.
Our results suggest that xoSNAT3 services as a SNAT involved in MT biosynthesis and pathogenicity in Xoo. We believe that our manuscript fits both the scope and quality of Journal of Environmental Research and Public Health and hope that it can be published.
Thanks for your consideration!
Sincerely yours,
Prof. Fengquan LIU
2022/12/1
Comments and Suggestions for Authors
Some observations to authors, included some mistakes and phrases and paragraphs less comprehensible.
ABSTRACT
Line 18: “…a SNAT related protein…”
Response: Thanks for the suggestion. We have changed it to “a SNAT protein”
INTRODUCTION
Line 41: “As in the case of vertebrates, MT also plays a key role as a master regulator in 41 plant growth and stress responses [12]”. The phrase is unclear, plants are not vertebrades. Omit: As in the case of vertebrates.
I suggest summarize information related to vertebrates and highlight information related to plant-microorganism interaction. MT is involved in plant growth regulation, evidence suggests that Xoo produces MT. Therefore, Xoo produce MT to manipulate plant metabolism? Why? or MT is a bacterial modulator? Any relationship with Quorum sensing?
Response: “As in the case of vertebrates” was removed in line 44.
Following the indication, the information regarding MT in humans was summarized in lines 38-42.
Additionally, a new discussion was added in lines 387-391 to explain the potential role of MT in plant-bacterial interaction.
MATERIALS AND METHODS
Line 191: “…(EcoRI):…”
Line 216: “ … using 2-ΔΔCt method…”
Line 220: “…on 50 mL NB. All…”
Response: Thanks for the suggestions. Mentioned mistakes were correctedin lines 201, 226 and 230.
RESULTS
Line 250: “…Figure 1A). Interestingly…”
Line 277. The concentration of intermediates did not showed statistical differences (Figure S1).
Line 282. Table 1 is absent
Line 284: “…Phylogenetic analysis…”
Response: Thanks for the suggestion. Mentioned mistakes were corrected in lines 258, 274, 285 and 289. “Table 1” was replaced by “Figure 3".
Line 292: the figure 3 corresponds to phylogenetic reconstruction, however it is difficult to understand due to the complex terms. The Xoo sequences should be highlighted.
Information about sequence alignment and nucleotide substitution model is absent.
Response: Thanks for the suggestion. We have present the xoSNATs and sequence alignment (Figure S1) and rewrite the “Phylogenetic analysis” section. Red dots highlighting Xoo sequences were added in Figure 3. A description about how the phylogenetic tree was constructed was added in the Caption of Figure 3.
Section 3.2 “xoSNAT3 is the first reported SNAT from Xanthomonas” is difficult however it is difficult to understand due to the complex terms. I suggest author should use simplified terms instead codes from NCBI.
Response: Many thanks, we have taken this suggestion, Mentioned mistakes were corrected in lines 326.
Section “3.3. xoSNAT3 is probably located in Xoo mitochondria” is not comprehensible. Since bacteria lack mitochondria. The authors should revise the results and rewriting.
Response: Many thanks, we have taken this suggestion. Mentioned mistakes were corrected in lines 333-335.
Line 323: What kind of cells? Author should be specific.
Response: Many thanks, we have specific this cells (tobacco leaves cells)
Figure 6. Letters of statistical significance among treatments are absent.
Response: Many thanks, we have added letters of statistical significance among treatments.
Figure 7. I suggest authors should include representative photographs of damaged tissue and biofilms. Letters of statistical significance among treatments are absent.
Response: Many thanks, we have taken this suggestion. Mentioned mistakes were corrected in lines379 and Figure 7.
Sections 3.2 and 3.2 are problematic since I did not perceive continuity between such sections. The phylogenetic reconstruction is not clear. The transient expression is not clear. The employed terms are difficult to follow. I suggest the authors would rewrite the sections.
Response: Many thanks, we have taken this suggestion. Mentioned mistakes were corrected in lines 289-325.
DISCUSSION.
The authors speculate that biosynthesis of MT in Xoo may be similar to vertebrates. However, additional evidences are not shown. I suggest authors discuss about MT biosynthesis in diverse prokaryotes, including both pathogens and no pathogens.
Response: Many thanks for the suggestion. As I have mentioned before, the research on bacterial MT synthesis pathway has not been reported yet. And I’m very embarrass, because there has not relevant research article of prokaryotes MT biosynthesis that could offer me to discuss.
The phrase “Our result suggested xoSNAT3 was located in mitochondria, which is a similar to that observed in animal cells. However, whether MT biosynthesis pathway in Xoo is similar to the one in animal cells needs further exploration” is very problematic. I did not understand the respective interpretation.
Response: Many thanks. We have changed it to “xoSNAT3 was located in mitochondria in tobacco cells”. As we known, it hard to find the location of xoSNAT3 in Xoo cell. And we have discuss why” MT biosynthesis pathway in Xoo is similar to the one in animal cells”, based on previously and our work.
Line 406. The figure S3 is the first time was mentioned. The information is unclear.
Response: Many thanks, we have deleted this figure.
CONCLUSION
The evidence suggests that Xoo produces MT as a part of pathogenicity, the host rice produces MT. However, a discussion about the relationship about MT biosynthesis considering the plant-pathogen interaction is necessary. Also a discussion about similarities between biosynthetic pathways from eukaryotes and prokaryotes is required. In my own perspective, the proposed model for biosynthesis of MT in Xoo is not well supported by current experimental evidence.
Response: Many thanks, we have deleted this figure. Moreover, we have modified this section by following your suggestion.
The evidence about the localization of xoSNAT was not considered in conclusion.
The manuscript includes figures but not tables. Information and discussion about Figures S1 and S2 is absent.
Response: Many thanks, we have taken this suggestion.

Reviewer 2 Report
1. Line 77 - what does "hazardous" mean?
2. Line 126 and 194 "sequencing" - please identify what platform or method
3. line 175 "PCR fusion method" simple description or cite source
4. Figure 7 A and B lesion length and OD values are replicated can be calculated statistically rather than just standard error bars to conclude that it is indeed involved in biofilm formation.
Author Response
Dear Editor,
We are resubmitting the manuscript titled "Molecular cloning and characterization of a serotonin N-acetyltransferase gene, xoSNAT3, from Xanthomonas oryzae pv. oryzae" to international Journal of Environmental Research and Public Health, in the section of Environmental Health, in the special Issue "New Strategies for the Environment-Friendly Management of Plant Diseases.
Comments from reviewers of the previous submission were greatly appreciated, and our manuscript has been modified accordingly. Attached are the response to reviewers.
Rice bacterial blight caused by Xoo is a major rice disease and serves as a model system for studying disease resistance in plants. The molecular mechanisms underlying the direct effect of melatonin on plant pathogenic are very complex. Serotonin N-Acetyltransferase (SNAT) is one of the key rate-limiting enzymes in melatonin (MT) biosynthesis. Here, we have identified SNAT related protein (xoSNAT3) in Xoo. xoSNAT3 was localized on mitochondria and has serotonin catalytic activity. Knock out xoSNAT3 impaired MT biosynthesis, and showed 14.06% less virulence and 26.07% less biofilm formation than the wild type strain.
Our results suggest that xoSNAT3 services as a SNAT involved in MT biosynthesis and pathogenicity in Xoo. We believe that our manuscript fits both the scope and quality of Journal of Environmental Research and Public Health and hope that it can be published.
Thanks for your consideration!
Sincerely yours,
Prof. Fengquan LIU
2022/12/1
Reviewer 2
Comments and Suggestions for Authors
- Line 77 - what does "hazardous" mean?
Response: Many thanks, we have changed it to “dangerous”. Mentioned mistakes were corrected in line 79.
- Line 126 and 194 "sequencing" - please identify what platform or method
Response: Many thanks, we have modified it by provide the platform information. Mentioned mistakes were corrected in line 130.
- line 175 "PCR fusion method" simple description or cite source
Response: Many thanks, we have provide this PCR description. Mentioned mistakes were corrected in lines 181-186.
- Figure 7 A and B lesion length and OD values are replicated can be calculated statistically rather than just standard error bars to conclude that it is indeed involved in biofilm formation.
Response: Many thanks, we have modified this section by following your suggestion. Mentioned mistakes were corrected in lines 368-370, 377, 379.
Submission Date
26 October 2022
Date of this review
22 Nov 2022 21:26:35

Round 2
Reviewer 1 Report
The authors claim that Xanthomonas oryzae pv. oryzae (Xoo) is able to biosynthesize melatonine, as part of virulence strategy, throught putative Serotonin N-Acetyltransferase (SNAT) gene. Authors should deeper the phylogenetic reconstruction and functional annotation of bacterial SNAT gene, since the related results and discussion are not well supported. The evolutive context of SNAT gene should get insights about novel mechanistic process of MT biosynthesis. I think it could be the novelty instead reporting % similarity to known SNAT genes. However, important sections in Material and Methods are absent, like confocal microscopy and western Blot.
Also, some errors and missing information are described below.
Line 36: …in both plant and bacterial kingdom…
Line 39: …Immunity [6-9]. MT is a super radical scavenger,….
Line 47:…recently identified in Arabidopsis. Despite…
Line 93: …Phylogenetic analysis… à Model of nucleotide substitution was not indicated.
Line 94: …The latest version of GNAT family amino acid sequences from Xanthomonas were downloaded from the NCBI website… à accesión number is absent. Authors should indicate the accession numbers of sequences employed in this paper.
Line 132: …bioinformatics analysis of xoSNAT3…. à Parameters for bioinformatic predictions are absent.
Sections 2.6 and 2.7: PJET1.2, PCV-eGFP –N1 vector. à the authors should provide maps of genetic vectors. Authors should provide details about tobacco genetic transformation.
Line 205: “…EcoRI and BamHI to release the cloned frag…
Line 226: “…The expression data, in terms of cycle treshold (Ct), were collected…” Authors should indicate test for statistical signficance.
Line 229: “2.11 Growth determination” à The authors should indicate measured parameters, and test for statistical signficance.
Line 241: Thirty leaves were treated with each strain. The.
experiments were conducted three times. à Authors should indicate test for statistical signficance.
Line 258: … Xoo synthetized MT, endogenous MT was…
Line 270: …The intermediate metabolites were extracted…
“In order to identify Xoo SNAT, 24 full-length proteins belonging to GNAT super- family (pfam00583), which were obtained from Xoo protein database
(https://www.ncbi.nlm.nih.gov/), were submitted to BLAST search” à My comment: The functional annotation of Xoo proteome was not indicated. Essential details like accession numbers, genome accession, parameters employed in protein database, sequence alignment, are absent. The use of SNAT sequences of vertebrades are not well justified. The phylogenetic reconstruction is unclear if it is based on DNA or protein sequences. However, nucleotide or aminoacid susbstitution models are absent. The phylogenetic tree reveals details about evolutive context of a series of sequences, instead similarity.
Line 302: “Phylogenetic reconstruction of genes involved in MT biosynthesis in Xoo”.
Lines97 and 304: The phylogenetic tree is based on máximum likelihood or neighbor joining?
Line 311: The size of xoSNAT3-GST and GST proteins is unknown. The gel 4ª should contain a mark indicating expected size protein.
Line 312: “… confirmed by western blot….” à The western blot methodology is absent.
Line 318: significant differences are absent.
Line 314: were 709.98 μgM -à The units are unknown. The authors should discute about if parameters are consistent with previously described to known SNAT.
Line 330: “…confocal microscope…” Methodology of confocal microscopy is absent.
Line 356: The text of figure 6 is not comprehensible. EXample, what means PX099? The paragraph related to Figure 6 should be more explainable.
Line 354: “The results suggested â–³xoSNAT3 is critical for MT biosynthesis in Xoo”. The sentence is not comprehensible since â–³xoSNAT3 is a mutant strain.
Line 379: Figure 7A and 7B. Authors should indicate test for statistical signficance. Pie figure of 7D is absent.
Line 390: “The role of MT in Xoo quorum sensing… My comment: authors should avoid comment about gaps in knowledge.
Line 398: …speculates that the synthetic pathway of MT in P. fluorescens RG11… This conclusión is not well supported. Aditional references are required for supporting it. The SNAT gene from RG11 was not included. The authors should enrich discussion: similar evolutive context of SNAT genes would be related to MT biosynthetic process. Or may be not.
Line 406: “However,
Line 407: “…ous TDC that involving…” What means TDC?
Lines 416: “…However, whether MT biosynthesis pathway in Xoo is similar to the one in animal cells needs further exploration..” My comment: authors should avoid comment about gaps in knowledge.
Line 422: “270 μgM…” Units did not correspond to Michaelis-Menten constant.
Lines 434: “…oxSNAT3…” It means xoSNAT3..?